# Membrane water for probing neuronal membrane potentials and ionic fluxes at the single cell level

M.E.P. Didier[1], O.B. Tarun[1], P. Jourdain[2], P. Magistretti[2] & S. Roke[1]

Neurons communicate through electrochemical signaling within a complex network. These signals are composed of changes in membrane potentials and are traditionally measured with the aid of (toxic) fluorescent labels or invasive electrical probes. Here, we demonstrate an improvement in label-free second harmonic neuroimaging sensitivity by ~3 orders of magnitude using a wide-field medium repetition rate illumination. We perform a side-by-side patch-clamp and second harmonic imaging comparison to demonstrate the theoretically predicted linear correlation between whole neuron membrane potential changes and the square root of the second harmonic intensity. We assign the ion induced changes to the second harmonic intensity to changes in the orientation of membrane interfacial water, which is used to image spatiotemporal changes in the membrane potential and $K^+$ ion flux. We observe a non-uniform spatial distribution and temporal activity of ion channels in mouse brain neurons.

[1] Laboratory for fundamental BioPhotonics (LBP), Institute of Bioengineering (IBI), and Institute of Materials Science (IMX), School of Engineering (STI), and Lausanne Centre for Ultrafast Science (LACUS), École Polytechnique Fédérale de Lausanne (EPFL), 1015 Lausanne, Switzerland. [2] Laboratory of Neuroenergetics and cellular dynamics, Brain Mind Institute, École Polytechnique Fédérale de Lausanne (EPFL), 1015 Lausanne, Switzerland. Correspondence and requests for materials should be addressed to S.R. (email: sylvie.roke@epfl.ch)

Neuronal signaling occurs through rapid changes in the membrane potential that originate from a redistribution of ionic charges across the plasma membrane. In mammalian neurons, $Na^+$, $Cl^-$, and $K^+$ are mainly responsible for regulating the membrane potential, and of these, $K^+$ is considered to be the most important[1]. The movement of these ions through specific ion channels leads to an adjustment of the membrane potential.

A membrane potential arises as a consequence of an imbalance in the ionic strength between two aqueous solutions that are separated by an impermeable membrane. This static difference in the membrane potential is described by the Nernst equation[2]. In neurons the membrane potential is further regulated by ion channels, whose functions were discovered in experiments by Hodgkin, Huxley and Katz[3,4]. By varying the extracellular concentration of $K^+$ ions around the axons of a squid, while electrically recording the change in resting membrane potential Hodgkin, Huxley and Katz demonstrated the essence of the membrane resting potential regulating mechanism. Ion channels present in the membrane regulate the imbalance in ionic strength and thus change the membrane potential[1]. This behavior is modeled by the Goldman, Hodgkin and Katz (GHK) equation that relates the membrane potential ($\Delta\Phi_0$) to the concentration of ionic species ($K^+$, $Na^+$, $Cl^-$) and the permeability ($P$) of ion channels:

$$\Delta\Phi_0 = \frac{kT}{e}\ln\frac{P_K[K^+]_{out} + P_{Na}[Na^+]_{out} + P_{Cl}[Cl^-]_{in}}{P_K[K^+]_{in} + P_{Na}[Na^+]_{in} + P_{Cl}[Cl^-]_{out}} \quad (1)$$

For neurons, the potential gradient is mainly controlled by $K^+$ ions[1] and changing it from 5 to 50 mM results in a less negative membrane-resting potential. Such a reduction in membrane potential magnitude is called depolarization. Using for Eq. (1) standard literature values ($[K^+]_{in} = 145$ mM, $[Na^+]_{in} = 15$ mM, $[Mg^{2+}]_{in} = 0.5$ mM, $[Ca^{2+}]_{in} = 70$ mM, $[Cl^-]_{in} = 10$ mM and $[K^+]_{out} = 5$ mM, $[Na^+]_{out} = 145$ mM, $[Mg^{2+}]_{out} = 1$ mM, $[Ca^{2+}]_{out} = 2$ mM, $[Cl^-]_{out} = 110$ mM)[1], $\Delta\Phi_0$ changes from $-75.4$ mV ($[K^+]_{out} = 5$ mM) to $-33.3$ mV when $[K^+]_{out}$ is changed to 50 mM.

Mapping neuronal electrical activity in cells in vitro and in real time is a challenging task that follows two main approaches[3,5–8]: electrophysiological and imaging. The electrophysiological approach, with invasive patch-clamp techniques[7], is used to record whole-cell changes in membrane potential[4]. Patch-clamp measurements rarely exceed the time scale of several minutes and are limited to the soma (dictated by the size of the pipette and the neuronal structures). Recent schemes using chronic implants can reduce but not eliminate these effects[9].

The main imaging techniques involve calcium imaging[5,10,11], optogenetic approaches[6,12], and interferometric microscopy[13,14]. These techniques rely on measuring physiological changes in the cell that are indirectly correlated to neuronal activity, such as changes in refractive index[15,16], stress-induced birefringence[17] or changes in the one or two-photon fluorescence emission from $Ca^{2+}$-responsive labels[18,19]. Recent work with stimulated Raman spectroscopic imaging has shown promising results in relating spectral C–H stretch bands to membrane potential changes in erythrocite ghost cells, still with an empirical connection between the Raman shift amplitudes and the membrane potential[20]. Second harmonic generation (SHG) is an elastic two-photon scattering process in which two photons with frequency $\omega$ are transformed by a non-centrosymmetric or anisotropic material into one photon with twice the frequency ($2\omega$). Non-centrosymmetric crystals, interfaces[21], or a broken centrosymmetric orientational distribution of (dipolar) fluorescent molecules by an electric field can generate second harmonic (SH)

photons[22]. SH imaging is used to image non-isotropic polar structures such as collagen[23,24] and muscle myosin[25,26]. Additionally, fluorescent voltage sensitive dyes have been used together with resonantly excited SHG to image membrane potentials[27–30]. In these studies the optical response was correlated to the reorientation of the chromophore in the membrane[29]. Membrane potential sensitive chromophores can significantly alter the electric property of the membrane and modify the normal physiological behavior of the cell, can be instable and toxic[31] thereby severely limiting the applicability of the methods[32].

As SHG is an inherently weak optical process, label-free imaging of neurological membrane potentials has not yet been achieved; an improvement in detection efficiency of several orders of magnitude was foreseen to be needed[23]. Recently, we demonstrated high throughput wide-field second harmonic imaging of glass/water interfaces[33] for which we improved the throughput of a SH microscope to a factor of ~5000, enabling the measurement of interfacial water that is oriented by the presence of surface charges inside a borosilicate micro-capillary. We used the interfacial water response to construct maps of the electric surface potential by relating the interfacial electrostatic field ($E_{DC}$) and corresponding surface potential ($\Phi_0$) via[34–36]:

$$I(2\omega, x, y, t) \sim I(\omega, x, y, t)^2 \left| \chi_s^{(2)} + \chi^{(3)'} f_3 \int E_{DC}(0, x, y, z)dz \right|^2$$
$$\cong \left| \chi_s^{(2)} + \chi^{(3)'} \Phi_0(x, y) \right|^2, \quad (2)$$

where $\chi_s^{(2)}$ is the surface second-order susceptibility and $\chi^{(3)'}$ an effective third-order susceptibility of the aqueous phase that primarily depends on oriented water in the electric double layer, and $f_3$ is an interference term that takes the value 1 for a transmission experiment[36,37]. Given the low photo toxicity of the high throughput microscope[38], it is possible to use the same approach, namely the reorientation of water in the electric double layer of a membrane to map membrane potentials in living neurons optically and label-free.

Here, we demonstrate the possibility of label-free imaging of the electrical neuronal activity of primary cortical neurons in vitro employing the membrane water as a probe for membrane potentials. We first characterize the SH throughput for neuroimaging, showing that label-free SH images of neurons can be recorded on 600 ms timescale. We then revisit the experiments of Hodgkin, Huxley and Katz[3] by detecting the membrane potential of primary cortical neurons during a $K^+$-induced depolarization. Using the nonlinear optical response of interfacial water and nonlinear optical theory we convert the SH images into membrane potential and $K^+$ ion flux maps. While the average temporal response agrees with the theoretically expected membrane potential changes from the GHK equation and the electrophysiological recordings obtained with the same protocol, the images show clear spatiotemporal fluctuations across the neuron. Within the framework of the GHK description, we observe a non-uniform density distribution and temporal activity of permeant ion channels. Being able to non-invasively probe membrane potentials and ion flux in active neurons, opens up avenues to understand signaling mechanisms, either on the single neuron level or in a larger network.

## Results

**Label-free high-throughput second harmonic neuroimaging.** To characterize the response for the neurons, and our imaging system for neuroimaging, we measured spectral responses, and computed the Michaelson image contrast and the signal to noise

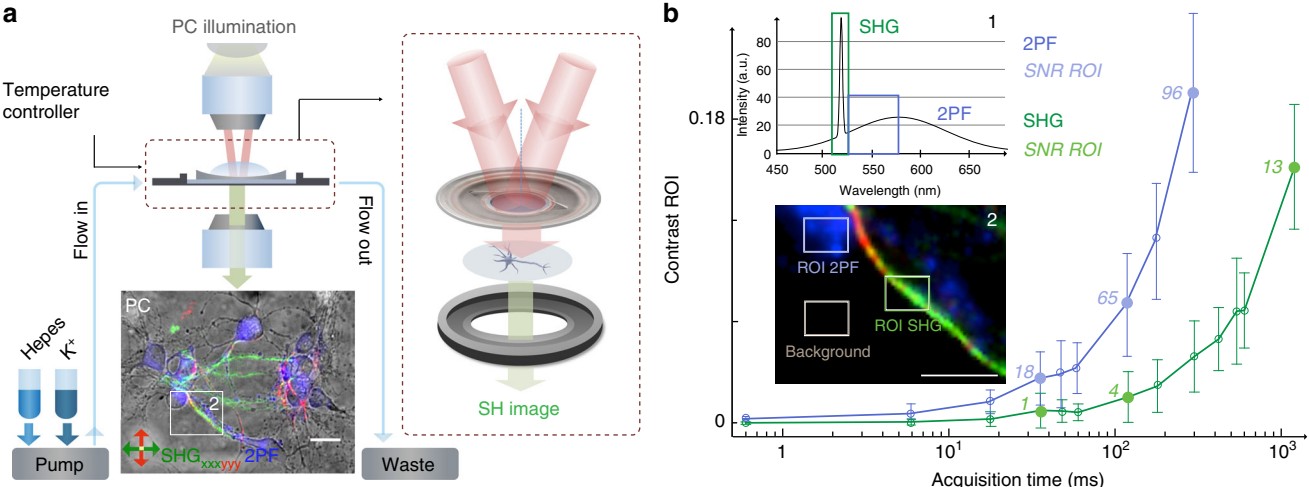

**Fig. 1** Protocol and imaging configuration with second harmonic throughput for neuroimaging. **a** Imaging configuration and protocol with a composite image of 17 days in vitro (DIV) primary cultured cortical mouse neurons. The neurons are provided with a constant flow of solution via a peristaltic perfusion system and SH imaging is done in a wide-field double beam transmission geometry. The composite image, is composed of a phase contrast (PC), label-free SH (in green and red) and 2PF, and the scale bar is 20 μm. The double head red and green arrows in the image represent the orientation of the respective incoming and analyzed polarization for SH signal. The emitted spectrum is composed of a second harmonic (SH) component and an endogenous two-photon fluorescence (2PF) contribution, as shown in inset 1 in panel **b**. **b** Michaelson contrast with signal to noise ratio (SNR) values in italic, calculated from the region of interests (ROIs) indicated by the rectangles in the zoomed inset 2 (scale bar, 10 μm) for the SH and 2PF images. Error bars were calculated from the standard deviation of the intensity values from the respective ROIs and drawn as vertical error bar at each data point. The insets show, 1 the emission spectra from a neuron with the SHG response (narrow peak, being at the exact double frequency of the 1030 nm fundamental) and the 2PEF response (a broad band at lower frequencies than the SH frequency) clearly visible, and 2 the zoomed inset from the image in panel **a** used to calculate the contrasts and SNRs

ratio (SNR) from cultured neurons. Figure 1a shows an illustration of the experiment. The neurons are kept in an open chamber under a constant flow with a buffer solution containing HEPES, maintained at 37 °C (see Methods for details) and are checked for mature electric activity prior to each experiment by observing K$^+$-induced swelling/shrinking using phase contrast imaging[39] (see Supplementary Note 1 and Supplementary Figure 1). The optical nature of the measured response was verified by recording the emitted spectrum. Figure 1b inset 1 shows the spectrum emitted from a single neuron, showing a SH peak on top of a broad endogenous two-photon fluorescence (2PF) response. Note that there is also some three photon fluorescence emission. For the imaging, the SH response is selected by using a 10 nm bandwidth filter centered around 515 nm, and the 2PEF response is selected by using a 25 nm bandwidth filter centered around 550 nm. A composite image of in vitro cortical neurons is also shown in Fig. 1a, using phase contrast (PC) imaging (black and white scale), label-free SH imaging mode (green, red, using different polarization directions of the beams indicated with xxx and yyy) and endogenous two-photon fluorescence (2PF, blue). The SH and 2PEF images were recorded with a fluence of 2.15 mJ/cm$^2$ and a peak intensity of 12.79 GW/cm$^2$. Figure 1b shows the Michaelson contrast (a definition can be found in the method section) for SH (green curve), and endogenous 2PF (blue curve) for the ROIs defined in Fig. 1b inset 2. SNRs are displayed as numbers next to the data points. It can be seen that images with an SHG SNR of 4 can be recorded with an acquisition time of 100 ms. Comparing this to literature[40] where label-free SH neuro-images were recorded with scanning microscope systems with acquisition times of 120 s fluences of 340 mJ/cm$^2$, and peak intensities of 1700 GW/cm$^2$, the improvement in throughput is more than three orders of magnitude.

**Whole-cell patch clamp and SHG measurements**. To verify the validity of Eq. (2) and make the connection with Eq. (1), we next

adjust the cell membrane potential by increasing the extracellular concentration of K$^+$ ions from 5 mM (normal concentration) in the extracellular buffer solution to 50 mM. We then record the voltage difference induced by this K$^+$ depolarization in two ways: by means of an electrophysiological patch clamp and with SH imaging. We subsequently verify the connection between the two predicted by Eq. (2).

Figure 2a shows a phase contrast (PC) image of a patched neuron (17 days in vitro) used for the experiment. Figure 2b shows the obtained electrophysiological recording in whole-cell current clamp mode. Figure 2c shows a composite PC and SH image recorded at the beginning of the experiment (before depolarization), and Fig. 2d displays the normalized square root of the SH intensity recorded under identical conditions and with the same protocol from a single neuron coming from the same culture.

It can be seen (Fig. 2b) that the resting membrane potential of the neuron $\Delta\Phi_0$ which corresponds to the difference in surface potentials of the two membrane leaflets) changes from −65 to −34 mV. According to Eq. (2), the square root of the SH intensity should scale with the membrane potential. Inspecting Fig. 2d, it can be seen that the changes in the square-rooted SH intensity have a strong temporal correlation with the changes in the potential recording from the electrophysiology technique. Indeed, plotting $\sqrt{I(2\omega)}$ versus $\Delta\Phi_0$ in Fig. 2e results in a linear dependence. Thus, it appears that also in the case of electrical activity of live neurons, the nonlinear optical response of water can be used as a marker of electrostatic potentials, just like it does at the glass/water interface[33]. That this is so can be understood from the following reasoning: a non-resonant SH experiment reports on all dipolar molecules that are not centrosymmetrically distributed, and each dipolar molecule has a response that is of the same order of magnitude[21]. The non-centrosymmetric distribution of dipolar molecules arises from chemical interactions at the interface (first term in Eq. 2) and the interaction with

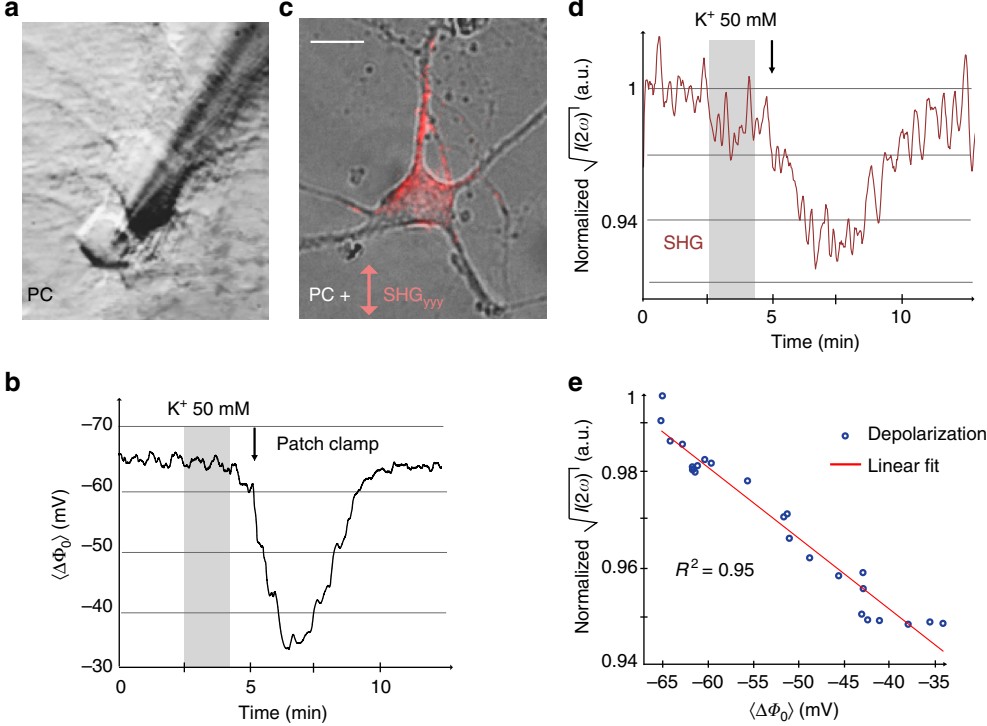

**Fig. 2** Second harmonic and electrophysiological response during $K^+$-induced depolarization. **a** Primary cortical neurons in culture undergoing a $K^+$ depolarization, patched (**a**, **b**) and probed with second harmonic generation (SHG) (**c**, **d**). **b** Whole-cell current clamp potential recording, the gray shadow area represents the time lapse application of $K^+$ solution and the arrow shows when the neurons are undertaking the solution. **c** Composite image of the phase contrast (PC) and second harmonic (SH) image of a cultured neuron, scale bar 20 µm. **d** The normalized square-rooted spatially averaged SH response of the single neuron depicted in **b**. The square-rooted spatially averaged SH intensity follows the temporal trend of the patch-clamp experiment. **e** Linear correspondence between the changes of the square-rooted SHG intensity and the membrane potential from patch-clamp recording

the interfacial electrostatic field (second term in Eq. 2). For an aqueous interface the most abundant dipolar species by far is water, even in the interfacial double layer region, where the orientational distribution of dipolar molecules is changed by the interfacial electrostatic field. On top of that, the measured SH intensity depends quadratically on the number density of dipolar species, enlarging even further the contribution of water molecules. For this reason, a surface in contact with water exhibits a SH response that arises predominantly from the reorientation of water[35–47]. This reasoning is general, and the SH response of water has been used to determine the membrane potential of liposomes[37], and free-standing lipid membranes in aqueous solutions that were subject to different external potentials[48]. Coupling these results from physics to neuroscience and using the fact that electrical signaling occurs through the working of ion channels, together with the assumption that the working of these ion channels can be modeled accurately by the GHK equation (Eq. 1) we arrive at the conclusion that the SH response from interfacial water can be used to determine the membrane potential of neurons.

**Spatiotemporal mapping of $\Delta\Phi_0$ and $K^+$ ion efflux.** Having recorded the dependence of electrophysiological membrane potential and SH intensity during $K^+$ induced depolarization of neurons and having assigned the response to water, we now turn to extracting label-free spatiotemporal variations in the membrane potential from the recorded SH images. To do so, we first need to update Eq. (2) to reflect better the structural composition of neurons and then apply assumptions that allow us to extract spatiotemporal membrane potential information during the time frame when the $K^+$ concentration is varying. As neurons are

composed of a cytoskeleton that is constructed from non-centrosymmetric microtubules[41,42] (MTs) it is necessary to take them into account as well as the presence of a lipid membrane (mem). Equation (2) thus becomes:

$$I(2\omega, x, y) \sim C(x,y)\left|\chi_{\text{MT}}^{(2)}(x,y) + \Delta\chi_{\text{mem}}^{(2)}(x,y) + \chi^{(3)'}\Delta\Phi_0(x,y)\right|^2$$

(3)

where $C(x,y)$ is a constant that converts counts to response units (m$^4$ V$^{-2}$) and includes $I(\omega,x,y)$, the intensity profile of the incoming beams. $\chi_{\text{MT}}^{(2)}(x, y)$ is the second-order susceptibility of the microtubules in the cytoskeleton, $\Delta\chi_{\text{mem}}^{(2)}(x, y) = (\chi_{s1}^{(2)} - \chi_{s2}^{(2)})(x, y)$ is the net surface second-order susceptibility of the membrane that is composed of oppositely oriented leaflets s1 and s2, and $\Delta\Phi(x,y)$ is the membrane potential difference. To determine $\Delta\Phi_0(x,y)$ from the reorientation of water by the interfacial electrostatic field, we need to determine solutions for $\Delta\chi_{\text{mem}}^{(2)}(x, y)$, $C(x,y)$, and $\chi_{\text{MT}}^{(2)}(x, y)$. Previous SH imaging and scattering experiments on asymmetric free-standing lipid membranes[37] and liposome solutions have shown that $\chi^{(3)'} = 10.3 \times 10^{-22}$ m$^2$/V$^2$, and $\Delta\chi_{\text{mem}}^{(2)}(x, y) = \Delta\chi_{\text{mem}}^{(2)} = 5 \times 10^{-24}$ m$^2$/V. To determine $C(x,y)$, we use the GHK equation and two known values of $\Delta\Phi_0$, assuming that at steady state the membrane potential distribution across the neuron is constant. At steady state conditions, when the extracellular concentration of $K^+$ is constant at $[K^+]_1 = 5$ mM (with $\Delta\Phi_{0,1} = -75.4$ mV) and at $[K^+]_2 = 50$ mM (with $\Delta\Phi_{0,2} = -33.3$ mV). Comparing the average intensity over 10 frames ($\Delta t$) and subtracting the two resulting time-averaged SH intensities to eliminate the contribution of every

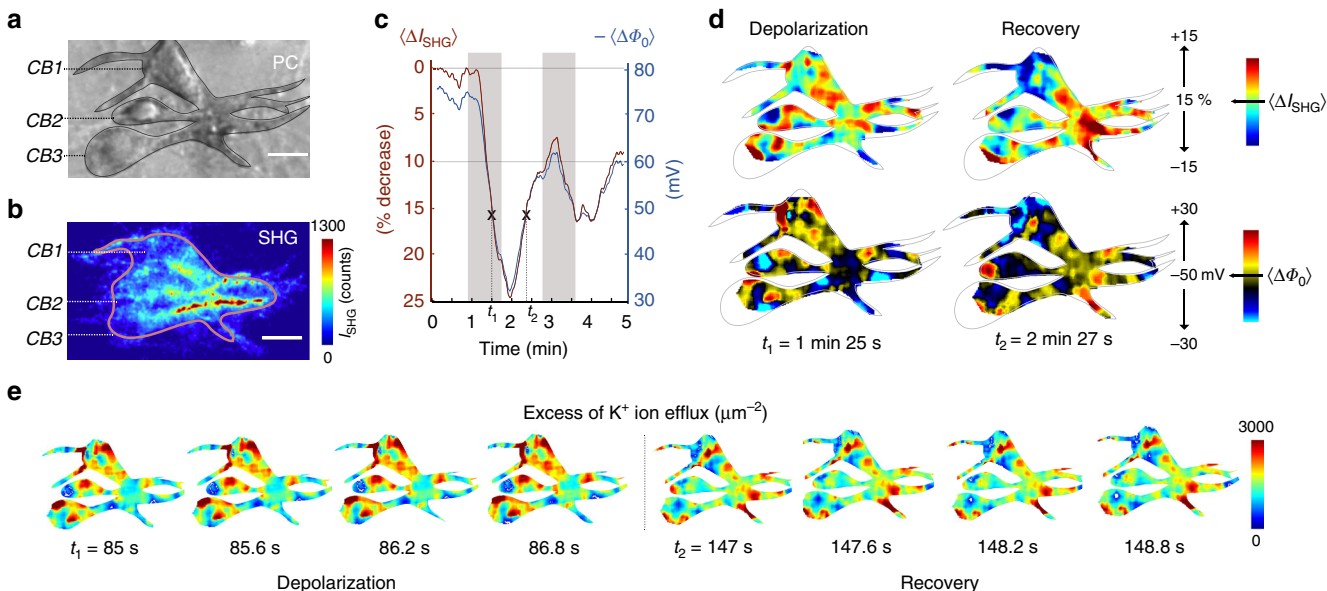

**Fig. 3** Spatiotemporal imaging of membrane potentials and ion efflux by means of water. **a**, **b** PC (**a**) and corresponding label-free SH image (**b**) of cortical neurons, 15 days in vitro, *CB* states for Cell Body. Three CBs are seen in the image denoted by *CB1*, *CB2* and *CB3* and a bundle of several processes. The colored line in **b** indicates the area in which the membrane potential calculations were made. Scale bar 20 μm. **c** Spatially averaged SH intensity (left axis) changes as a function of time during application cycles of 50 mM $K^+$ enriched extracellular solution. The average membrane potentials derived from the computational procedure are shown on the right axis. The time windows were $K^+$ enriched solution is applied are highlighted by the gray zones. **d** Top row: Images of the percentile change in the SH response at two different times ($t_1$ and $t_2$) in the membrane depolarization cycle, indicated by crosses in **c**. The bottom images display the corresponding membrane potential maps. **e** Excess of $K^+$ ion efflux ($μm^{-2}$) showing snapshots of spatiotemporal dynamic changes with 600 ms intervals during the continuous depolarization cycle from $t_1 = 85$ to 86.8 s and during the recovery period from $t_2 = 147$ to 148.8 s. Supplementary Movie 1 contains the recorded videos of the presented data

$X^{(2)}$ value, we can determine $C(x,y)$ via:

$$C(x,y) = \frac{\left\langle \sqrt{I(x,y)(\Delta\Phi_{0,1})} \right\rangle_{\Delta t1} - \left\langle \sqrt{I(x,y)(\Delta\Phi_{0,2})} \right\rangle_{\Delta t2}}{x^{(3)'}(\Delta\Phi_{0,1} - \Delta\Phi_{0,2})} \quad (4)$$

To estimate compatible (but not unique) $\chi_{MT}^{(2)}(x,y)$ values we use the same assumption and calculate back $\chi_{MT}^{(2)}(x,y)$, with known values for $\Delta\Phi_{0,1}$, $\chi^{(3)'}$, and $C(x,y)$. The resulting spatially varying range of values for $\chi_{MT}^{(2)}$ is $-2\times10^{-22} < \chi_{MT}^{(2)} < 0.1 \times 10^{-22}$ $m^2$/V, is in agreement with expectations, as they are an order of magnitude larger than an oriented interfacial monolayer[37].

Finally, the spatio-temporal fluctuations of the calculated membrane potential $\Delta\Phi(x,y,t)$ are determined by substituting the previously calculated $C(x,y)$, $\chi_{MT}^{(2)}(x,y)$ and $\Delta\chi_{mem}^{(2)}$ values into Eq. (3).

**Spatiotemporal heterogeneities in ion channel activity.** Figure 3 shows the results of the above procedure. Figure 3a shows a PC image of neurons displaying three somas and processes, highlighted by the black lines (somas), open circles (dendrites) and axons (filled circles). Figure 3b displays a SH image recorded over a time span of 120 s prior to the membrane depolarization experiment. Upon changing the extracellular concentration of $K^+$ from 5 mM to 50 mM (during the time intervals from $t = 1$ min to $t = 1$ min 45 s and $t = 2$ min 45 s to $t = 3$ min 40 s, highlighted by the gray areas) the spatially averaged SH intensity changes, dropping by 25%. When the flow is reversed back to 5 mM $K^+$ the SH intensity recovers, although not fully due to the short waiting time, and then drops back down when the extracellular $K^+$ concentration is switched back to 50 mM. The computed space-averaged membrane potential value $\Delta\Phi_0$ is shown on the right axis and follows the trend of the SH intensity. The average value

agrees well with the electrophysiological measurements of Fig. 2c. Figure 3d shows pairs of images displaying the percentile changes in the SH intensity (top) and the computed membrane potential maps (bottom) for an average membrane potential of −50 mV. Spatial fluctuations across the three neurons are visible in the different images. These fluctuations are, within the framework of the GHK equation, due to local and temporal fluctuations in ion channel distribution or activity. This agrees with recent super resolution fluorescent microscopy studies of fixed cells[43]. During the depolarization, the membrane potential reaches more positive values in the somas, while during recovery the neurites display more positive membrane potential values. Hence, different parts of the cells (soma or neurites) are active at different points of the depolarization cycle. We can display these changes more clearly by considering that the membrane potential change is determined mainly by the $K^+$ activity[1]. Assuming that $K^+$ ions are the only potential determining ions, the ion flux ($\Delta Q/\Delta t$) can be computed[44] using the relationship between the difference in charge ($\Delta Q$) of two frames (separated by $\Delta t$), the membrane capacitance ($C_m$), and the membrane potential difference ($\Delta\Phi_0$), $\Delta Q = C_m\Delta\Phi_0$. The capacitance of the membrane has a typical value[45] of 1 μF/$μm^2$. Knowing $\Delta\Phi_0$ from our SH measurements, we can construct ion flux maps. Figure 3e shows snapshots of the excess $K^+$ ion efflux for a few 600 ms intervals during the continuous depolarization cycle from $t_1 = 85$ to 86.2 s and during the recovery period from $t_2 = 147$ to 148.2 s. Supplementary Movie 1 contains the recorded videos of the presented data that show the full membrane potential changes and ion efflux for all the recorded frames. These videos show that there is not only a non-uniform distribution and density of permeant ion channels, but also a different temporal activity of $K^+$ ionic efflux.

Summarizing, we propose the nonlinear optical response of membrane water as a mechanism for probing membrane potentials and ion fluxes label-free. We have used the endogenous response of

interfacial water to image the sub-cellular and time dependent response of living cortical mammalian neurons to a potassium-enriched solution. The enrichment in $K^+$ concentration initiated a change in the resting membrane potential of the neuron. Electrophysiological recordings confirmed this well-known response. The temporal square-rooted response of the SH intensity that originates from the oriented water consistently followed the same trend confirming expectations from nonlinear optical theory. We then analyzed the recorded images and constructed maps of the membrane potential as well as the $K^+$ ion flux. We observed temporal variations and spatial variations in the membrane potential and $K^+$ ion flux. These changes are due to a non-uniform distribution of ion channels within the neurons. Different parts of the cells were active at different parts of the depolarization cycle. Thus, within the framework of the GHK description, there is not only a non-uniform distribution and density of permeant ion channels, but also a different temporal activity. We should note here, though, that we used the GHK equation as an approximation, which undoubtedly has simplified the real complexity of a living system. Several aspects of electrical pulse propagation have been reported to suggest a higher complexity involving a coupling between electrical and mechanical responses[49–51] and a more complicated energy dissipation[52] than what would be expected from simple Ohmic heating. Future experiments, using multiple modalities, such as SH imaging of water in combination with techniques such as iSCAT[53], digital holography[54], and stimulated Raman scattering[20] will potentially allow one to understand the coupling between electrical and mechanical activity related to neuro-signaling.

Being able to non-invasively probe membrane potentials and ion fluxes, opens up avenues for understanding the signaling mechanism inside single neurons or within a larger network. For the present work our method is limited to the mapping of membrane resting potentials and processes occurring on time scales of ~0.1–1 s. While the present 600 ms recordings are still longer than the duration of action potentials, future improvements to the optical layout and noise reduction may bring the recording time down to the 1 ms time scale. The possibility of simultaneous multisite membrane potential or ion flux recordings allows one to resolve the local neuronal network activity; a prerequisite for comprehensive studies of the relationships between spatiotemporal activity patterns, neuronal network development, and information processing. Measurements on tissues are also possible, and endoscopic human clinical applications may become within reach one day, given the straightforward relationship between membrane potential and the probed water, the absence of toxic changes, or the need for genetic modifications.

## Methods

**Cell culture preparation**. Primary cultures of cortical neurons were prepared from E17 OF1 mice embryos. The average plated density of cells was 15,000 cells/cm² per coverslip. Cultured neurons were maintained at 37°C in a humidified atmosphere of 95% air/5% $CO_2$ and were used for experiments after 14 days in vitro.

**Whole-cell patch-clamp recordings**. Single whole-cell recordings (current clamp) were performed using a EPC10 amplifier (HEKA Elektronik, Lambrecht, Germany) controlled by PatchMaster software. Patch pipettes with tip resistance of 5–7 MΩ were pulled and fire polished prior to the measurements. Intracellular solution (Intracellular buffer, Fivephoton biochemicals) contained (mM): 140 NaCl, 10 $Na_2HPO_4$, 2 $KH_2PO_4$, 3 KCl, pH 7.3 ± 0.1, with an osmolarity of 290 ± 10 mOsm.

**SHG and endogenous 2PEF imaging system**. The imaging system is composed of a Yb:KGW amplified laser (Pharos Light Conversion) with 200 kHz, 1036 nm, 168 fs laser pulses and described in detail in ref.[33]. The pulse energy delivered to the sample is 36 μJ and shaped into a circular illumination spot size with a FWHM of 150 μm in the sample plane, reaching a fluence of 2.15 mJ/cm² and peak intensity of 12.79 GW/cm². The laser beam illuminates a spatial light modulator (SLM, Phase Only Microdisplay 650–1100 nm, fill factor of 93%, with a 1920 × 1080-pixel resolution and a 8 μm pixel pitch from HOLOEYE Photonics) under normal incidence. Used as a reflective diffraction grating, the SLM generates a gray scale hologram image of a rotatable slit pattern. The pattern is filtered in the Fourier plane of a lens placed after the SLM, such that only the 1st and −1st orders are retained. These beams are then imaged onto the sample plane using an Olympus LUMPLFLN ×60 1 NA water immersion objective such that it is illuminated with two wide-field beams with a tunable opening angle that can be set by varying the grating spacing period on the SLM. The polarization of the two incoming beams is controlled by a zero-order half-wave plate (Thorlabs, WPH05M-1030). The generated SH light was collected in the forward direction using a high NA Olympus, LUMFLN 60 × 1.1NA objective and analyzed with an analyzer placed in the collection path after a 515 nm band pass filter (Omega Optical, 10 nm bandwidth). The endogenous 2PEF response was recorded using a 50 nm-broad bandpass filter (BP550/25) filter. Imaging was performed with a gated detection on an electron-multiplying intensified charge-couple device (EM-ICCD, PiMax4, Princeton Instruments). The microscope also integrates a path for phase contrast imaging with white light. The SH $K^+$-induced depolarization experiments were repeated 7 times on different cultures under the same experimental conditions.

**Sample cell and perfusion system**. Coverslips with plated neurons were inserted in a Quick Change Imaging Chamber from Warner Instruments (Series 40 RC-41LP). This chamber provides a maximum liquid content of 3 mL, a laminar flow across the chamber and an open bath. A constant flow of extracellular solution with 140 mM NaCl, 5 mM KCl, 3 mM $CaCl_2 \cdot 2H_2O$, 2 mM $MgCl_2 \cdot 6H_2O$, 5 mM glucose and 10 mM Hepes (denoted as HEPES solution) was used at a rate of 1 mL/mn associated with a suction speed of 1.2 mL/mn. After 2 minutes the HEPES solution was switched to a solution enriched in $K^+$ with 93 mM NaCl, 50 mM KCl, (denoted as $K^+$ solution). The solution was switched back (syringe pump Harvard Apparatus PHD 2000) to HEPES after 2 min and the wash out lasted for 5 min.

**Image processing and data analysis**. The raw imaging data were collected from LightField (Princeton Instruments), stored as a TIF stack and exported to Matlab (R2016b). The specific set of data mentioned in the paper is composed of a stack of 500 frames with a temporal acquisition of 0.6 s per frame. The Michaelson contrast of Fig. 1b was calculated for each SH image and plotted as a function of acquisition time. The Michelson contrast is defined as $C_{Michelson} = \frac{(I_{SH,max} - I_{SH,min})}{(I_{SH,max} + I_{SH,min})}$. With $I_{SH,max}$, the maximum SH intensity and $I_{SH,min}$ the minimum SH intensity found in the image. Denoising was done by subtracting an offset, applying a Fast Fourier Transform filter along the temporal dimension, and applying thresholding and a median filter in the spatial dimension for each frame (kernel = 3).

## Data availability
The data that support the findings of this study are available from the corresponding author upon reasonable request.

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

## Acknowledgements

This work was supported by the Julia Jacobi Foundation and the Swiss National Foundation (grant number 200021_146884). The authors would like to thank E. Ruchti, E. Gasparetto and M. Wirth for providing the neuronal cultures, B.S. Sermet for help with the patch-clamp measurements and P. Marquet and I. Allaman for discussions.

## Author contributions

M.E.P.D. performed experiments; P.J. and P.M. provided the patch-clamp experimental protocol, and were involved in discussions throughout the project, M.E.P.D., O.B.T., and S.R. interpreted the data and wrote the manuscript. S.R. conceived and supervised the work.

## Additional information

**Competing interests:** The authors declare no competing interests.

