## [Peer Review File · Nature Communications]

Reviewers' comments:

Reviewer #1 (Remarks to the Author):

Didier et al present a study of neuronal membrane potential imaging using water orientation as a probe. The interest of this study is very high since the technique offers label-free imaging capabilities of neuronal activities.

However, the scientific evidences advanced by the authors does not allow to publish the study. Several points question the conclusions of the authors. The authors do not prove that the signal is only second harmonic generation since two and three photons fluorescence can be detected in the same time on the same channel. The authors do not prove that the signal variation come from water orientation only.

Moreover

- The potential orients the water molecules at the vicinity of the membrane, which allow the SHG imaging. A zoom around a dendrite with a high quality would have been good to see the origin of the signal. Also at different time.
- If the signal comes from the water orientation around membranes, why is there a strong signal from the nucleus? There are so many membranes into the nucleus?
- How can you prove that is SHG of water? Many others molecules can orient as well.
- Can you even prove that is SHG? The detection filtering is based on a band pass filter at 515/10nm. With the power laser used and the very efficient detection, a three-photon fluorescence signal can be generated and detected on this channel. A temporal trace of the fluorescence around 515nm (530 or 500nm) or using a delay with the triggered camera would have been good to be sure of the signal origin. A K⁺ flux can quench the auto fluorescence and explains the decrease of the signal.
- In the manuscript, some references are not well numbered
- Authors do not explain what Michelson SH image is
- Fluorescence method is not explain

The origin of the signal is not convincing and an important number of supplementary experiments must be done to answer to the points above. In this case, the paper must be resubmitted.

Reviewer #2 (Remarks to the Author):

This is an exciting paper that reports a new approach for the non-invasive measurement of transmembrane potentials. It should be published. I have one minor comment which I would like the authors to think about and possible to add some discussion in the paper:

While the Hodgkin and Huxley (HH) model is successful in modeling the observed electrical phenomena, it does not account for several other aspects of nerve pulse propagation. For example, it is known that a mechanical displacement propagates alongside the electrical pulse [1,2], and that the net heat release of an axon during a nerve pulse is less than expected from the Ohmic heating of an electrical cable [3].

Indeed it was experimentally found that pressure waves propagate in such a system [4] which was theoretically explained by the existence of a surface-localized fractional wave [5]. So I think it would be important to discuss the possibility that the SHG signal does pick up some of the side effects (i.e. change of membrane density and change of temperature).

[1] G H Kim, P Kosterin, A L Obaid, and B M Salzberg, *Biophysical Journal* 92 3122-3129 (2007).

[2] A El Hady and B B Machta, *Nature Communications* 6, 6697 (2015).

[3] I Tasaki, P M Byrne, *Jpn. J. Physiol.* 42 805-813 (1992).

[4] J Griesbauer, S Bössinger, A Wixforth, and M F Schneider, *Phys. Rev. Lett.* 108, 198103- 198103 (2012).

[5] Julian Kappler et al *PHYSICAL REVIEW FLUIDS* 2, 114804 (2017)

Reviewer #3 (Remarks to the Author):

The authors in this study applied a wide-field second harmonic (SH) imaging technique to map membrane potential and potassium ion flux in neurons upon chemical depolarization. The manuscript provides good and detailed theory and calculation of potential correlation between SH signal and membrane potential. Authors used high potassium buffer to depolarize neurons and observed SH signal changes that correlate with the potassium-induced depolarization. Using this

signal change, membrane potential was calculated and mapped. Furthermore, using different calculation model, authors mapped the potassium ions during depolarization and re-polarization based on the same SH signal change. I recommend publication of this novel method under the condition that the following concerns are addressed.

1. Figure 3: Authors applied 50 mM potassium buffer twice, first for 0.45 min and second for 0.95 min. Although it is promising to see that SH intensity shows the trend of recovery in-between the two treatments, the level of intensity drops for the two treatments are not at the similar level. Is there a supporting evidence showing the change of membrane potential is indeed the same as shown by the SH intensity change or the computed membrane potential?
2. Figure 3: Based on SH intensity traces (both from figure 2d and figure 3c), it seems like there is a long delay for the SH intensity to recover during membrane repolarization. For example, in Figure 2, at 10 min, the membrane potential should have returned to resting potential according to Figure 2b. If assuming neurons from Figure 2b and Figure 2d have the similar response curve, the SH signal returns to baseline at ~ 12.5 min. 2.5 min of recovery time is a very slow kinetics, which will interfere with imaging of fast spiking.
3. Figure 3c: why hasn't membrane potential returned to the baseline ~ 80 mV? Was there any cytotoxicity from the procedure?
4. Figure 3e: I think potassium ion mapping is an interesting observation and, considering the experimental designs, reflect the meaning of the SH signal change better. However, it is a very sudden change of the flow of the whole manuscript. I am confused of whether SH signal should reflect membrane potential or potassium ion distribution, since membrane potential is not controlled solely by potassium ions. Moreover, control experiments to determine the distribution of potassium channels to check the correlation between the mapped potassium ion changes and channels are necessary to make the conclusion more solid and meaningful.

Minor

1. Although potential clinical applications were suggested, it is not clear how this can reach clinical application based on the results from this study because the experiments were all done in cultured neurons, and not even brain slice imaging were attempted.
2. The time for potassium treatments is described in $t=1$ min to $t=1.45$ min and from $t=2.45$ min to $t=3.40$ min in page 8, and $t_1 = 85$ s to 86.2 s and during the recovery period from $t_2 = 147$ s to 148.2 s in page 9. This unit inconsistency was also found in figure 3. Should make them consistent.

Detailed response to the reviewer's comments

We thank the referees for commenting on our manuscript. The original comments are given in *italic* and our answers are in normal font.

Reviewer #1 (Remarks to the Author):

Didier et al present a study of neuronal membrane potential imaging using water orientation as a probe. The interest of this study is very high since the technique offers label-free imaging capabilities of neuronal activities.

However, the scientific evidences advanced by the authors does not allow to publish the study. Several points question the conclusions of the authors. The authors do not prove that the signal is only second harmonic generation since two and three photons fluorescence can be detected in the same time on the same channel. The authors do not prove that the signal variation come from water orientation only.

We have added spectra that prove the SH nature of the response (Figure 1). The bandwidth of the filter used for SHG is 10 nm, neatly fitting around the SH response and cutting out the 2- and 3PF.

The origin of the SH response is attributed to water because:

It is well known that a surface in contact with water exhibits a SH response that arises from the reorientation of water. The reason is that in a non-resonant SH experiment all dipolar molecules contribute to the SH response, and each molecule has a response of the same order of magnitude. For an aqueous system the most abundant dipolar species by far is water, even in the interfacial double layer region, where the orientational distribution of dipolar molecules is changed by the interfacial electrostatic field. For example, in a 100 mM solution of solute s 1 in every 550 molecules belongs to the solute. In addition, the surface SH response is coherent and scales with the quadratic number difference, so that the ratio of SH intensities arising from the water and the solute is $(550/1)^2=302500$, (assuming the orientational distributions are similar).

The SH intensity from the interfacial water can be described by Eq. 2 that relates the surface potential to the interfacial water response (see e.g. Ong et al. 1992). This type of equation is used in second harmonic scattering measurements to retrieve the interfacial potential, and the interfacial nonlinear optical response of water was used to measure the membrane potential of liposomes in solution (Lütgebaucks et al. 2016). In addition, very recently, SH imaging measurements of free standing lipid membranes (surrounded by electrolyte solution) were performed and showed explicitly that the SH response arises from interfacial water, and that this response allows to describe a change of membrane potential using Eq. 2 (Tarun et al. 2018). The method and origin of the SH signal are thus general and work for every aqueous system.

In addition, in neuroscience the prevailing insights regarding electrical activity is that (Purves, Neuroscience 2004) electrical signaling occurs through the working of ion channels. The working of ion channels is modeled by the GHK equation (Eq. 1). This means that electrostatic gradients occur in the vicinity of the ion channels (due to current/field interactions).

Thus, within the framework of the above assumptions, which are general and reasonable, water can be assigned as the primary source of the SH response imaged here as a function of K^+ induced membrane resting potential changes.

In addition, we have recently performed experiments on voltage gated ion channels in free standing lipid membranes that confirm the origin of water. This data is unpublished but we would be willing to share preliminary results if the editor / referee so desires.

Moreover

- The potential orients the water molecules at the vicinity of the membrane, which allow the SHG imaging. A zoom around a dendrite with a high quality would have been good to see the origin of the signal. Also at different time.

The signal from water molecules arises from a depth of ~ 1 nm or less. With our current optical resolution (380 nm) we would not see any difference in the image (except in the value of the integrated intensity).

- If the signal comes from the water orientation around membranes, why is there a strong signal from the nucleus? There are so many membranes into the nucleus?

There are membranes around the nucleus and they have ion channels.

- How can you prove that is SHG of water? Many others molecules can orient as well.

Please see the above answer to the first question. We have added further information about this reasoning on page 1.

- Can you even prove that is SHG? The detection filtering is based on a band pass filter at 515/10nm. With the power laser used and the very efficient detection, a three-photon fluorescence signal can be generated and detected on this channel. A temporal trace of the fluorescence around 515nm (530 or 500nm) or using a delay with the triggered camera would have been good to be sure of the signal origin.

Please see the answer to the first question. We can assign the response to SHG from the spectrum which we have added to the new figure 1 (it is exactly the double of the input spectrum). In addition, changing the delay between both arms results in a vanishing SH signal.

A K⁺ flux can quench the auto fluorescence and explains the decrease of the signal.

The spectral data shows this is not the case; we have added spectra to the data in fig. 2d showing amplitude changes in the SH spectral response.

- In the manuscript, some references are not well numbered

We have changed this.

- Authors do not explain what Michelson SH image is

We have provided a definition of the Michelson contrast (see materials and methods section).

- Fluorescence method is not explain

We have updated the description in the materials and methods section.

The origin of the signal is not convincing and an important number of supplementary experiments must be done to answer to the points above. In this case, the paper must be resubmitted

We hope the presented new data and answers are convincing.

Reviewer #2 (Remarks to the Author):

This is an exciting paper that reports a new approach for the non-invasive measurement of transmembrane potentials. It should be published. I have one minor comment which I would like the authors to think about and possible to add some discussion in the paper:

While the Hodgkin and Huxley (HH) model is successful in modeling the observed electrical phenomena, it does not account for several other aspects of nerve pulse propagation. For example, it is known that a mechanical displacement propagates alongside the electrical pulse [1,2], and that the net heat release of an axon during a nerve pulse is less than expected from the Ohmic heating of an electrical cable [3].

Indeed it was experimentally found that pressure waves propagate in such a system [4] which was theoretically explained by the existence of a surface-localized fractional wave [5]. So I think it would be important to discuss the possibility that the SHG signal does pick up some of the side effects (i.e. change of membrane density and change of temperature).

*[1] G H Kim, P Kosterin, A L Obaid, and B M Salzberg, *Biophysical Journal* 92 3122-3129 (2007).*

*[2] A El Hady and B B Machta, *Nature Communications* 6, 6697 (2015).*

*[3] I Tasaki, P M Byrne, *Jpn. J. Physiol.* 42 805-813 (1992).*

*[4] J Griesbauer, S Bossinger, A Wixforth, and M F Schneider, *Phys. Rev. Lett.* 108, 198103- 198103 (2012).*

*[5] Julian Kappler et al *PHYSICAL REVIEW FLUIDS* 2, 114804 (2017)*

We thank the reviewer for this interesting comment. Indeed, given the complexity of a biological system it seems unlikely that a simplistic model like that of HH would capture 100 % of the ongoing physics, chemistry, and biology. One way of tackling to co-existence of mechanical deformations alongside the electrical changes would be to perform linear holography or interferometric light scattering in conjunction with the SHG (assuming that the spatial resolution would be sufficient). This would indeed be a very interesting avenue for future research.

We have added a paragraph at the end of the manuscript discussing such effects and possibilities (page 12).

Reviewer #3 (Remarks to the Author):

The authors in this study applied a wide-field second harmonic (SH) imaging technique to map membrane potential and potassium ion flux in neurons upon chemical depolarization. The manuscript provides good and detailed theory and calculation of potential correlation between SH signal and membrane potential. Authors used high potassium buffer to depolarize neurons and observed SH signal changes that correlate with the potassium-induced depolarization. Using this signal change, membrane potential was calculated and mapped. Furthermore, using different calculation model, authors mapped the potassium ions during depolarization and re-polarization based on the same SH signal change. I recommend publication of this novel method under the condition that the following concerns are addressed.

1. Figure 3: Authors applied 50 mM potassium buffer twice, first for 0.45 min and second for 0.95 min. Although it is promising to see that SH intensity shows the trend of recovery in-between the two treatments, the level of intensity drops for the two treatments are not at the similar level. Is there a supporting evidence showing the change of membrane potential is indeed the same as shown by the SH intensity change or the computed membrane potential?

The drops are at different levels since the neuron has not recovered yet, and the degree of depolarization is high.

This response is known as the refractory period and is actually quite reproducible, also for different cultures if one uses the same protocol. There may be small differences for different neurons, however, but not in the general trend. Our neuroscience collaborators have identified these responses to be quite uniform and representable by the Goldman equation (Purves, D. *Neuroscience. Sunderland (2004)*).

2. Figure 3: Based on SH intensity traces (both from figure 2d and figure 3c), it seems like there is a long delay for the SH intensity to recover during membrane repolarization. For example, in Figure 2, at 10 min, the membrane potential should have returned to resting potential according to Figure 2b. If assuming neurons from Figure 2b and Figure 2d have the similar response curve, the SH signal returns to baseline at ~ 12.5 min. 2.5 min of recovery time is a very slow kinetics, which will interfere with imaging of fast spiking.

This would perhaps be a concern for future studies. At this moment we cannot yet image fast spiking. This is why we focused on membrane resting potentials in this study.

3. Figure 3c: why hasn't membrane potential returned to the baseline ~ 80 mV? Was there any cytotoxicity from the procedure?

This is because the neurons have not recovered fully yet. During the experiment they are not supplied with food (to prevent random firing). As the depolarization is quite strong and for a relatively long time, it could also be that some fraction of the ion channels have been depleted, so that it would take relatively longer for the neuron to return to the resting state.

4. Figure 3e: I think potassium ion mapping is an interesting observation and, considering the experimental designs, reflect the meaning of the SH signal change better. However, it is a very sudden change of the flow of the whole manuscript. I am confused of whether SH signal should reflect membrane potential or potassium ion distribution, since membrane potential is not controlled solely by potassium ions. Moreover, control experiments to determine the distribution of potassium channels to check the correlation between the mapped potassium ion changes and channels are necessary to make the conclusion more solid and meaningful.

The SH signal measures potential. The potential distribution relates to the current flow according to a straightforward conversion (assuming we know the capacitance of the membrane). We assume the main current is from K^+ ions, since they are considered the potential determining ions (following neuroscience literature, Purves, D. *Neuroscience. Sunderland (2004)*).

Minor

1. Although potential clinical applications were suggested, it is not clear how this can reach clinical application based on the results from this study because the experiments were all done in cultured neurons, and not even brain slice imaging were attempted.

This study indeed was aimed at single neurons, because it represents more clearly the underlying physics and localized changes on the single cell level. It would be more difficult to resolve ionic flux / potential changes in individual cells in a tissue. Probing the tissue, however, would be much easier because the intensities are generally much stronger. Indeed, to move on to a clinical path, one needs to use tissues as an intermediate. The most straightforward clinical application in our opinion would be the use of an endoscope applied to spinal cord recordings.

2. The time for potassium treatments is described in $t=1$ min to $t=1.45$ min and from $t=2.45$ min to $t=3.40$ min in page 8, and $t_1 = 85$ s to 86.2 s and during the recovery period from $t_2 = 147$ s to 148.2 s in page 9. This unit inconsistency was also found in figure 3. Should make them consistent.

In Fig. 3, potassium is applied from 1 min to 1 min 45 s. The time lapse image shots in Fig. 3e are taken from $t_1=85$ s = 1 min 25 s during the depolarization period, which is contained between 1 min = 60 s and 1 min 45 s = 105 s. The Fig.3e during the first recovery period, from 1 min 45 s (=105 s) to 2 min 45 s (=165 s), are taken at 147 s, which is also contained in the recovery period.

REVIEWERS' COMMENTS:

Reviewer #1 (Remarks to the Author):

The authors provide new datas and answered to my concerns. I find it suitable for publication now

Reviewer #2 (Remarks to the Author):

I am happy with the changes made to the paper as reaction to minor comments.

Reviewer #3 (Remarks to the Author):

The authors have addressed my concerns.